# How Human-like Behavior of Service Robot Affects Social Distance: A Mediation Model and Cross-Cultural Comparison

**DOI:** 10.3390/bs12070205

**Published:** 2022-06-22

**Authors:** Linyao Li, Yi Li, Bo Song, Zhaomin Shi, Chongli Wang

**Affiliations:** 1School of Economics and Management, Chongqing University of Posts and Telecommunications, Chongqing 400065, China; lisalinyaoli@hotmail.com (L.L.); shizhaomin796@163.com (Z.S.); s200701040@stu.cqupt.edu.cn (C.W.); 2Post-Doctoral Station of Business Administration, Fudan University, Shanghai 200433, China; song@shnu.edu.cn; 3Institute of Tourism, Shanghai Normal University, Shanghai 200234, China

**Keywords:** human-like behavior, service robot, social distance, perceived competence, perceived warmth

## Abstract

Previous studies on the human likeness of service robots have focused mainly on their human-like appearance and used psychological constructs to measure the outcomes of human likeness. Unlike previous studies, this study focused on the human-like behavior of the service robot and used a sociological construct, social distance, to measure the outcome of human likeness. We constructed a conceptual model, with perceived competence and warmth as mediators, based on social-identity theory. The hypotheses were tested through online experiments with 219 participants from China and 180 participants from the US. Similar results emerged for Chinese and American participants in that the high (vs. low) human-like behavior of the service robot caused the participants to have stronger perceptions of competence and warmth, both of which contributed to a smaller social distance between humans and service robots. Perceptions of competence and warmth completely mediated the positive effect of the human-like behavior of the service robot on social distance. Furthermore, Chinese participants showed higher anthropomorphism (perceived human-like behavior) and a stronger perception of warmth and smaller social distance. The perception of competence did not differ across cultures. This study provides suggestions for the human-likeness design of service robots to promote natural interaction between humans and service robots and increase human acceptance of service robots.

## 1. Introduction

Robots can be used to perform a series of complex actions [1]. A service robot performs service tasks for humans or devices [2]. It is an autonomous robot capable of interacting with people and completing specific service tasks [1]. The development of artificial-intelligence technology has popularized service robots, such as educational robots, therapeutic robots, and entertainment robots [3]. However, human acceptance of service robots is the main obstacle to popularizing service robots [4]. Since service robots have certain social attributes [5] and human-like characteristics that can encourage humans to treat service robots as social participants, human-like characteristics can influence the service effectiveness of robots [5,6]. The human-like characteristics of robots can effectively influence human attitudes toward robots [6]. Human acceptance of the human-like characteristics of robots promotes human acceptance of service robots [7], whereas human non-acceptance of the human-like characteristics of robots inhibits human acceptance of service robots [6].

However, scholars have different views on human acceptance of human-like robots. Some scholars believe that humans have positive emotions toward human-like robots and are more willing to deal with a robot that has more human-like features [7,8,9], while some other scholars believe that more human-like robots can cause fear and anxiety in people, decreasing their willingness to interact with the robot [6,10]. This paper focused on how the human likeness of a service robot affects human acceptance of it.

Human likeness refers to the degree to which a robot looks and behaves like a human [11]. Human likeness includes two categories: human-like appearance and human-like behavior [12,13]. Appearance describes the static aspects of the robot (look, sound, sense of touch, etc.) [14,15,16], while behavior describes the dynamic aspects of the robot (actions, expressions, emotions, etc.) [11,12]. To enhance the human likeness of the service robot, the designer would endow the service robot with more human characteristics. For example, the designer would make a robot’s face look like a human’s or add more human characteristics to its actions [13]. A few previous empirical studies have explored the human-like behavior of service robots (HLBR) [8]. However, this factor also has an important effect on human–robot interaction [8]. Therefore, this paper focused on the effects of HLBR on human acceptance of a service robot.

In previous studies, scholars used two types of constructs to measure human acceptance of a service robot: (i) psychological constructs, such as trust [16,17,18], likes [11], use intention [19], and satisfaction [20]; and (ii) sociological constructs, such as social distance [21]. Most studies have employed psychological constructs, while few have employed sociological constructs. However, it is important to examine the human acceptance of a service robot from a sociological perspective. The previous literature has shown that the social rules in people-to-people interactions apply to human–robot interaction [22], and robots can be viewed as social actors with specific behavioral patterns [23]. This paper focused on the sociological aspect of human beings’ acceptance of service robots (i.e., social distance). Social distance refers to the closeness of the relationship between the two individuals in people-to-people interactions [24]. The social distance between humans and service robots (SDHR) can measure the closeness of the relationship between humans and service robots [25]. Thus, SDHR can indicate human acceptance of a robot [21].

This paper examined how HLBR affects SDHR. This paper has two contributions. First, this paper extends the outcomes of HLBR. Concerning the effect of HLBR on human acceptance of a robot, previous studies examined only the attitudes [8] and likes [26] but not any other outcomes. This paper examined the effect of HLBR on SDHR. Second, this paper introduced perceived competence and perceived warmth as mediators. The mediating effect of HLBR and SDHR has not been studied in the previous literature. This paper employed the social-identity theory to explain the mediating effect of HLBR and SDHR.

In addition, previous studies have shown that cultural background can affect human responses to robots [27,28,29]. In the US and China, robots are widely used in various fields, including the service industry [30], such as Sony’s entertainment robot AIBO and Takara’s home-care robot TERA [27]. However, the two countries differ in their views on robots. Americans regard robots as assistants, while Chinese tend to regard robots as friends [31]. It is generally believed that the US is an individualist country, and China is a collectivist country [32,33]. Compared to individualism, interpersonal relationships are more intimate in the context of collectivism [34]. Social rules in interpersonal communication can also apply to human–robot interaction [22]. A cross-cultural study of human–robot interaction found that Chinese people have a higher sense of intimacy with robots than Americans [27]. Therefore, we validated our theoretical model with participants from two different cultures (China and the US).

## 2. Literature Review and Hypothesis Development

### 2.1. Human Likeness and Social Distance

In the process of human–robot interaction, humans would perceive social distance from the robot [25]. Social distance can be understood as the closeness between two individuals’ relationships [35]. SDHR is the result of the dynamic interaction between human attributes (gender, age, and the experience dealing with the robot) and robot attributes (appearance and interaction cues) [25]. Previous studies have found that humans naturally attribute human characteristics to non-human objects [14]. Consumers would spontaneously give human attributes to, for example, cars [36] or brands [37]. Human-like service robots have some characteristics of humans [4]. The higher the human likeness of the robot, the richer the human characteristics of the robot, and the stronger the human perception of the similarity between robot and human [38]. Perceived similarity can affect an individual’s perceived social distance; the higher the similarity, the smaller the social distance [39,40]. As an aspect of the robot’s human likeness, a higher HLBR can also lead to a smaller SDHR. Based on this, we proposed the following hypothesis:

**Hypothesis** **1** **(H1).***The higher the HLBR is, the smaller the SDHR*.

### 2.2. Human Likeness as well as Competence and Warmth

Anthropomorphism is the tendency to attribute human-like qualities to non-human objects [41,42]. The robot’s human-like appearance can promote the robot’s anthropomorphism [8]. Anthropomorphism can enhance human emotional attachment to non-human objects in service scenarios [43]. When humans interact with anthropomorphized robots, they may feel an affinity with the robots [44]. Warmth and competence are basic dimensions used to characterize others [45]. Human perception of the robot’s competence is related to the capabilities, intelligence, skills, and other characteristics of the service robot, while the human perception of the warmth of the robot is related to the caring, friendliness, sociability, and other characteristics of the service robot [8,45]. Anthropomorphism affects these two basic judgment dimensions [6,46,47]. Studies have found that if the robot is anthropomorphized by the HLBR, human perception of the robot’s competence may increase [6], and human perception of the warmth of the robot may become more positive [8,46]. Therefore, we can speculate that HLBR may affect human perceptions of the competence and warmth of the service robot. A service robot with a high human-like behavior should be considered more competent and warmer by humans. Based on this, we proposed the following hypotheses:

**Hypothesis** **2** **(H2).**
*The higher the HLBR is, the stronger the perceived competence.*


**Hypothesis** **3** **(H3).**
*The higher the HLBR is, the stronger the perceived warmth.*


### 2.3. Competence, Warmth, and Social Distance

Social distance reflects the consciousness of kind in human sociological attributes [25,48]. The social-identity theory holds that humans would categorize individuals based on social-categorization cues [45]. Human perceptions of the competence and warmth of robots would serve as social-categorization cues [49,50] and affect the results of categorizing the social groups of robots by humans [43]. The subjective categorization of inter-and intra-social groups affects social distance [39]. When humans regard other individuals as members of the same group, social distance tends to be smaller [40]. Therefore, we can speculate that the stronger the human perception of the competence and warmth of the robot, the smaller the SDHR would be. Based on this, we proposed the following hypotheses:

**Hypothesis** **4** **(H4).**
*The stronger the perceived competence is, the smaller the SDHR.*


**Hypothesis** **5** **(H5).**
*The stronger the perceived warmth is, the smaller the SDHR.*


### 2.4. Mediating Effects of Perceived Competence and Perceived Warmth

Studies suggest that humans tend to be attracted to human-like objects because of their conformity with humans [36,51]. Competence and warmth are the two universal dimensions of human-impression formation [45,49], accounting for almost 80% of human impressions of others [49]. The robot’s human likeness can significantly affect these two basic judgment dimensions [47]. Van Doorn et al. found that perceived competence and perceived warmth mediate the relationship between a robot’s human likeness and the service performance of the robot (such as customer satisfaction and loyalty) [52]. Kim et al. found that the human likeness of the service robot affects consumer attitudes toward the service robot indirectly through competence and warmth [8]. Social distance is a construct close to satisfaction and attitude [25]. Therefore, we speculated that perceived competence and perceived warmth might mediate the relationship between HLBR and SDHR. Based on this, we proposed the following hypotheses:

**Hypothesis** **6** **(H6).**
*Perceived competence mediates the relationship between HLBR and SDHR.*


**Hypothesis** **7** **(H7).**
*Perceived warmth mediates the relationship between HLBR and SDHR.*


## 3. Methods

### 3.1. Participants

This study recruited participants from different cultural backgrounds. Among them, the 219 Chinese participants were recruited from Credamo (www.credamo.com) [53,54], while the 180 American participants were recruited from MTurk (www.mturk.com) [17,55,56,57]. All the participants from Credamo were Chinese. The participants from MTurk were people from different countries, so we selected the US participants by setting three questions (Where were you born? Where do you live? Have you ever lived in a country other than the United States for more than three months?). The characteristics of the participants are shown in Table 1.

### 3.2. Procedure

The Chongqing University of Posts and Telecommunications Ethics Committee reviewed the experimental procedures (School of Economics and Management, Project-2022-0003). We employed two videos as stimuli, which were used by scholars in previous studies [8] and which present the scenes of humans talking with robots. In the two videos, the robots’ appearances were the same, and the questions humans asked of the robots were also the same, but the robots answered and acted differently. In the low-HLBR setting, the robot took no actions, and its language expression had no emotional characteristics. In the high-HLBR setting, the robot could move its hands, nod its head, and express strong emotions through language. The conversations in the videos were in English. The videos for the Chinese participants contained Chinese subtitles. We employed back-translation techniques. Two research assistants, who were native Chinese but excellent in English, translated the videos. One of them translated the English conversations in the videos into Chinese, while the other subsequently translated the Chinese version back into English. A professor of consumer behavior compared the back-translated English with the English conversations in the videos to ensure language equivalence. See Appendix A for the URL of the full videos.

The procedures for the Chinese and American participants were the same. The introductory language and questionnaire items were in Chinese and English for the Chinese and American participants, respectively. We employed the same back-translation process that we used when translating the video conversations, to ensure language equivalence. The participants were first informed of the purpose of the experiment and the ways in which the data would be used. Subsequently, they were randomly assigned to a setting (high HLBR vs. low HLBR) to watch the corresponding video. On-page time for watching the video was at least the total time of the video (31 s/38 s). Next, the participants answered two questions related to the video contents (What is the robot’s main job in the video? What type of robot is the robot in the video?) to check whether they had truly engaged with the experiment and understood the video correctly. The robots in the videos provided answers to both questions. Finally, the participants responded to the questionnaire (with attention-check questions inserted) based on the video contents. After completing the questionnaire, each Chinese participant was paid RMB 3 (about USD 0.45), while each American participant was paid USD 0.4. We excluded any unqualified participants (participants who wrongly answered any question related to the video contents, wrongly answered any question among the attention check questions, or selected the same answer for all questions, or whose answers showed obvious regularity). The data from the remaining participants were used for analysis.

### 3.3. Measurement

The questionnaire items were adopted from previous studies, and some of them were adapted to fit the service-robot settings. The measurement items and sources of each construct are shown in Table 2. All items were measured on a 7-point Likert scale, ranging from strongly disagree (1) to strongly agree (7). In addition to measuring the three constructs (perceived competence, perceived warmth, and SDHR) included in the hypotheses, we also measured anthropomorphism. This construct (anthropomorphism) reflects the perception of HLBR of the participants, and we used it to test whether our manipulation of HLBR was successful. During subsequent hypothesis testing, we treated the HLBR as a dummy variable (1 = high, 0 = low).

## 4. Results

### 4.1. Reliability and Validity

We included the Chinese and American participants in the reliability and validity analysis [56,57], using SPSS 23.0 and LISREL 8.80. As shown in Table 3, the Cronbach’s α value of each construct exceeded the acceptable cut-off point of 0.7. The fit of the CFA model (four factors) was acceptable (χ^2^ = 274.697, df = 84, χ^2^/df = 3.270, RMSEA = 0.076, SRMR = 0.052, NNFI = 0.977, CFI = 0.981, IFI = 0.981, RFI = 0.967, GFI = 0.916, and AGFI = 0.880). The standardized factor loadings of each item were between 0.701 and 0.930 and were significant (*p* < 0.001). The combined reliability (CR) of each construct was greater than 0.7, and the average variance extracted (AVE) was greater than 0.5. The square root of each AVE was greater than the correlation coefficient between the constructs, as seen in Table 4. The above results indicated that the reliability, convergent validity, and differential validity of all the constructs were acceptable [61,62]. In addition, the fit of Harman’s single-factor model (χ^2^ = 1824.681, df = 90, χ^2^/df = 20.274, RMSEA = 0.220, SRMR = 0.111, NNFI = 0.803, CFI = 0.831, IFI = 0.832, RFI = 0.795, GFI = 0.621, and AGFI = 0.494) was significantly worse than that of the four-factor model, indicating that one latent variable cannot be used to replace the four factors. Therefore, the effect of common-method bias was not an issue in this study.

### 4.2. Manipulation Check

The independent samples t-test on the scores of anthropomorphism (the participants’ perception of HLBR) under the two settings of HLBR (high vs. low) was performed using SPSS 23. For the Chinese participants (t = 3.685, *p* = 0.000, and M _high_ = 5.100 vs. M _low_ =4.384) and the American participants (t = 2.796, *p* = 0.006, and M _high_ = 4.631 vs M _low_ = 4.027), the between-group difference in HLBR (high vs. low) was significant, indicating that the manipulation was successful.

### 4.3. Hypothesis Testing

The direct and indirect effects were tested using the PROCESS (Model 4). HLBR (high = 1, low = 0) was the independent variable, SDHR was the dependent variable, and perceived competence and warmth were mediators. Gender (1 = male, 0 = female) and age (natural logarithm) were included in the model as control variables. The bootstrapped sample size was 5000. The results are shown in Table 5, Table 6, Table 7 and Table 8 and in Figure 1.

The results of hypothesis testing was consistent for the Chinese and American participants.

The direct effect of HLBR on SDHR was not significant; therefore, H1 was not supported. HLBR was positively related to perceived competence, supporting H2. HLBR was positively related to perceived warmth, supporting H3. The perceived competence was negatively related to SDHR, supporting H4. The perceived warmth was negatively related to SDHR, supporting H5.

Both confidence intervals corresponding to the two mediating effects (HLBR → perceived competence → SDHR and HLBR → perceived warmth → SDHR) do not contain 0, indicating a significant mediating effect, supporting H6 and H7. Perceived competence and perceived warmth fully mediated the proposed relationship, in the absence of the direct effect of HLBR on SDHR.

### 4.4. Cross-Cultural Comparison

As an extended analysis, we compared the differences between the Chinese and American participants in their scores on the constructs. The results are shown in Table 9. The Chinese and American participants differed significantly in the scores on anthropomorphism (the participants’ perception), perceived warmth, and SDHR, whereas they did not differ in their scores on perceived competence.

## 5. Discussion

Our mediation-effect model was supported in both groups (the Chinese and American participants), indicating that the mediating effects of perceived competence and perceived warmth are applicable across cultures (China and the US), to a certain extent. However, the two groups differed in some aspects, stemming from cultural differences.

First, the Chinese participants’ score on anthropomorphism was higher than that of the US participants. Cross-cultural studies have found that compared to Americans, Chinese people more strongly advocate animism, so Chinese people are more inclined to anthropomorphize robots [31]. Therefore, when faced with the same HLBR, the Chinese people more strongly anthropomorphized the service robot in the experiment.

Second, the Chinese participants’ score on perceived warmth was higher than that of the American participants. Since the external characteristics of service robots (such as small size and slow movement speed) are more in line with Eastern cultural preferences [27], people from Eastern cultural backgrounds might have a more positive attitude toward human-like robots and a higher evaluation of the robot’s cuteness and friendliness [27,63]. These characteristics are linked to the human perception of the warmth of the human-like robot [45]. As a result, the Chinese people had a stronger perception of the warmth of the human-like robot.

Finally, the Chinese participants’ score on SDHR was lower than that of the American participants. Previous studies have shown that Americans are high on individualism, while Chinese are high on collectivism [32,33,34]. Compared to individualists, collectivists have a more pronounced in-group preference and maintain a smaller social distance from in-group members [39]. Since the Chinese people tend to view robots as in-group members [64], they keep a smaller social distance from human-like service robots.

### 5.1. Theoretical Contributions

First, this paper extends the study on the outcomes of HLBR. Few previous empirical studies have examined HLBR, and the existing literature has mainly discussed HLBR related to customer attitude [8] and user preference [29]. This paper linked HLBR with SDHR and examined the outcome of HLBR from a sociological perspective.

Second, this paper extended the study on the antecedents of SDHR. Social distance is an important indicator of human acceptance of robots [21], but only a few empirical studies have explored the antecedents of social distance. The existing literature has studied only the effect of the robot’s language form on social distance [25]. This paper extended the antecedents of social distance to HLBR (including the conversation contents, voice intonation, and actions of the robots when they interact with humans) to investigate the effect of the robot’s behavioral characteristics on SDHR.

Third, this paper demonstrated that perceived competence and warmth mediate the relationship between HLBR and its outcome. Previous studies on the robot’s human likeness and human acceptance of the robot have found that mediators include social-interaction needs [43] and the sense of social presence [65]. The mediators (i.e., perceived competence and perceived warmth) identified in this paper are fundamental dimensions of social cognition [45], and they help us understand the mechanisms by which HLBR produces outcomes, from the perspective of human perception.

Fourth, this cross-cultural study tested the model’s universality, with participants from China and the US. This paper also found differences between the Chinese and American participants in their perception of service robots. These results complement cross-cultural studies on human attitudes toward robots [66,67].

### 5.2. Practical Implications

First, it is necessary to consider human-like behavior when designing service robots. Robots’ expressions, attitudes, and actions can enhance human acceptance of service robots. This paper responded to the debate on the necessity of the human-like design of robots [7,9,10,19]. The findings of our study suggest that investing resources in the design of HLBR is beneficial.

Second, designing the HLBR is conducive to humans’ acceptance of service robots. The findings of our study showed that the design of HLBR can enhance human in-group identification with service robots (represented by a smaller social distance), making service robots not only accepted by humans but also better integrated with human groups in a sociological sense [25,68].

Finally, the detailed design of HLBR, such as the conversation contents, specific voice intonation, and specific actions, should aim to promote individuals’ positive perceptions of the competence and warmth of service robots as the mediators of the relationship between a robot’s human-like behavior and service satisfaction [43,49,50].

### 5.3. Limitations and Future Research

First, the model in this paper did not include human characteristics. However, the existing studies showed that human characteristics are also important factors affecting SDHR. Future research may incorporate human characteristics into the model. Second, this paper used video as the stimuli, rather than stimuli generated by real contact between humans and robots, which may have led to different results than would be expected. Future research may conduct field studies to examine real human—robot contact. Third, the Chinese and American participants showed some differences in the perception of service robots, for multiple reasons that we discussed in the previous section. Future research may conduct a more nuanced cross-cultural analysis. Fourth, we selected participants based on their country of origin; therefore, selection bias may have affected the experimental results. Future research could improve the selection method.

## 6. Conclusions

This paper aimed to explore the effects of HLBR on SDHR. Based on social identity theory, we constructed a conceptual model with perceived competence and perceived warmth as mediating variables. We recruited participants from China (219) and the United States (180) to complete the online experiment. Through empirical testing, we found that high HLBR (vs. low) led to higher perceived competence and perceived warmth, which shortened SDHR.

## Figures and Tables

**Figure 1 behavsci-12-00205-f001:**
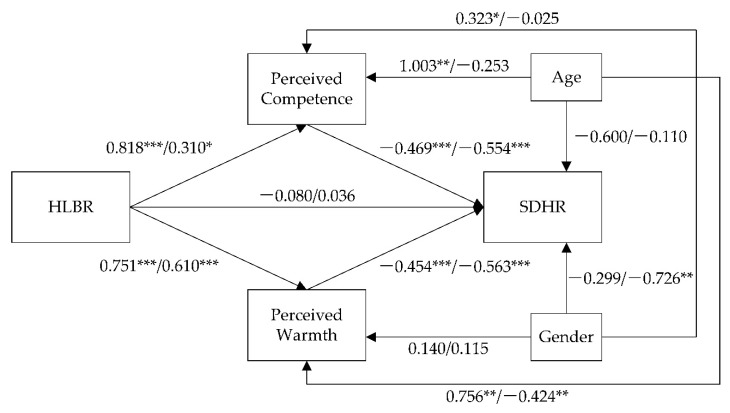
Results of mediation analysis. Note: Two coefficients on the path are China/US; * *p* < 0.05, ** *p* < 0.01, *** *p* < 0.001.

**Table 1 behavsci-12-00205-t001:** Descriptive statistics for the participants.

	Category	China (Frequency)	America (Frequency)
Gender	Male	100	99
	Female	119	81
Age	18–25	100	21
	26–30	56	52
	31–40	56	52
	41–50	2	36
	51–60	4	13
	More than 60	1	6
HLBR setting	High	127	93
	Low	92	87
Total		219	180

**Table 2 behavsci-12-00205-t002:** Measurement items of each construct.

Construct	Items	References
Anthropomorphism	How did you perceive the robot? (Machine-like or Human-like)	Bartneck et al., 2009 [51]
	How did you perceive the robot? (Fake or Natural)	
	How did you perceive the robot? (Unconscious or Conscious)	
SDHR	I think I have the potential to become friends with this robot.	Yilmaz et al., 2015 [58]; Joo et al., 2018 [59]
	I would like to keep in touch with this robot as a friend.	
	I would like to introduce this robot to others as my new friend.	
	I think this robot has the potential to be a dear friend to whom I can pour my heart out.	
Perceived Competence	The robot is capable.	Fiske et al., 2002 [60]
	The robot is efficient.	
	The robot is intelligent.	
	The robot is skillful.	
Perceived Warmth	The robot is friendly.	Fiske et al., 2002 [60]
	The robot is well-intentioned.	
	The robot is warm.	
	The robot is sincere.	

**Table 3 behavsci-12-00205-t003:** Results of reliability and validity analysis.

Factors	Items	Standardized Factor Loadings (λ)	T-Value	Cronbach’sα	Composite Reliability (CR)	Average Variance Extracted (AVE)
Anthropomorphism	AT1	0.811	18.666	0.845	0.848	0.650
	AT2	0.832	19.373			
	AT3	0.774	17.481			
SDHR	SDHR 1	0.904	23.162	0.940	0.941	0.799
	SDHR 2	0.930	24.334			
	SDHR 3	0.904	23.152			
	SDHR 4	0.835	20.314			
Perceived Competence	Comp1	0.829	19.477	0.877	0.880	0.647
	Comp2	0.787	18.056			
	Comp3	0.764	17.292			
	Comp4	0.835	19.710			
Perceived Warmth	Warm1	0.701	15.299	0.859	0.858	0.603
	Warm2	0.735	16.310			
	Warm3	0.863	20.560			
	Warm4	0.796	18.258			

Note: N = 399.

**Table 4 behavsci-12-00205-t004:** The mean, standard deviation, correlation coefficient, and AVE’s square root.

Variable	Mean	SD	1	2	3	4
1. Anthropomorphism	4.591	1.482	0.806	−0.664 ***	0.603 ***	0.556 ***
2. SDHR	3.691	1.713	−0.738 ***	0.894	−0.533 ***	−0.586 ***
3. Perceived Competence	5.056	1.147	0.685 ***	−0.583 ***	0.804	0.494 ***
4. Perceived Warmth	5.227	1.150	0.666 ***	−0.681 ***	0.552 ***	0.777

Note: Correlation coefficients between latent variables (from LISREL) are below the diagonal, diagonal values represent square root of AVE, and Pearson-correlation coefficients (from SPSS) are above the diagonal; *** *p* < 0.001.

**Table 5 behavsci-12-00205-t005:** Model test of mediation analysis (China).

Dependent Variable	Variable	β	SE	T	95% Confidence Interval		R^2^	F
					LLCI	ULCI		
Perceived Competence	Constant	1.163	1.095	1.062	−0.995	3.320	0.172	14.930 ***
	HLBR	0.818 ***	0.159	5.130	0.502	1.132		
	Gender	0.323 *	0.163	1.979	0.006	0.645		
	Age	1.003 **	0.336	2.983	0.083	1.666		
Perceived Warmth	Constant	2.617 **	0.860	3.044	0.922	4.312	0.186	16.413 ***
	HLBR	0.751 ***	0.125	5.997	0.504	0.998		
	Gender	0.140	0.128	1.093	−0.113	0.393		
	Age	0.756 **	0.264	2.863	0.236	1.277		
SDHR	Constant	10.368 ***	1.081	9.560	8.237	12.499	0.452	35.069 ***
	HLBR	−0.080	0.170	−0.471	−0.415	0.255		
	Perceived Competence	−0.469 ***	0.070	−6.665	−0.608	−0.330		
	Perceived Warmth	−0.454 ***	0.090	−5.062	−0.630	−0.277		
	Gender	−0.299	0.160	−1.872	−0.613	0.016		
	Age	−0.600	0.335	−1.792	−1.260	0.060		

Note: N = 219; LLCI = lower-level confidence interval, ULCI = upper-level confidence interval; * *p* < 0.05, ** *p* < 0.01, *** *p* < 0.001.

**Table 6 behavsci-12-00205-t006:** Model test of mediation analysis (US).

Dependent Variable	Variable	β	SE	T	95% Confidence interval		R^2^	F
					LLCI	ULCI		
Perceived Competence	Constant	5.768 ***	0.939	6.145	3.915	7.620	0.028	1.715
	HLBR	0.310 *	0.147	2.108	0.020	0.600		
	Gender	−0.025	0.151	−0.167	−0.324	0.274		
	Age	−0.253	0.259	−0.976	−0.764	0.258		
Perceived Warmth	Constant	5.891 ***	1.069	5.510	3.781	8.001	0.087	5.583**
	HLBR	0.610 ***	0.167	3.646	0.280	0.940		
	Gender	0.115	0.172	0.664	−0.226	0.455		
	Age	−0.424	0.295	−1.439	−1.006	0.158		
SDHR	Constant	9.643 ***	1.538	6.271	6.608	12.678	0.403	23.524***
	HLBR	0.036	0.224	0.160	−0.406	0.478		
	Perceived Competence	−0.554 ***	0.140	−3.971	−0.829	−0.279		
	Perceived Warmth	−0.563 ***	0.123	−4.594	−0.804	−0.321		
	Gender	−0.726 **	0.223	−3.254	−1.166	0.286		
	Age	0.110	0.383	−0.287	−0.646	0.865		

Note: N = 180; LLCI = lower-level confidence interval, ULCI = upper-level confidence interval; * *p* < 0.05, ** *p* < 0.01, *** *p* < 0.001.

**Table 7 behavsci-12-00205-t007:** Mediating-effect test (China).

		Effect	Boot SE	95% Confidence Interval	
				LLCI	ULCI
Direct effect		−0.080	0.170	−0.415	0.255
Indirect effect	Total	−0.724	0.146	−1.032	−0.457
	HLBR → PC → SDHR	−0.384	0.106	−0.605	−0.197
	HLBR → PW → SDHR	−0.341	0.100	−0.554	−0.169

Note: N = 219; LLCI = lower-level confidence interval, ULCI = upper-level confidence interval; PC: perceived competence, PM: perceived warmth.

**Table 8 behavsci-12-00205-t008:** Mediating-effect test (US).

		Effect	Boot SE	95% Confidence Interval	
				LLCI	ULCI
Direct effect		0.036	0.224	−0.406	0.478
Indirect effect	Total	−0.515	0.173	−0.865	−0.190
	HLBR → PC → SDHR	−0.172	0.089	−0.357	−0.006
	HLBR → PW → SDHR	−0.343	0.130	−0.637	−0.128

Note: N = 180; LLCI = lower-level confidence interval, ULCI = upper-level confidence interval; PC: perceived competence, PW: perceived warmth.

**Table 9 behavsci-12-00205-t009:** Comparison of variable means for Chinese and American participants.

Variables	China	US	T
Anthropomorphism	4.799	4.339	3.119 **
SDHR	3.281	4.190	−5.364 ***
Perceived Competence	5.088	5.018	0.620
Perceived Warmth	5.607	4.765	7.803 ***

Note: ** *p* < 0.01, *** *p* < 0.001.

## Data Availability

Not applicable.

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
