# Peer review of "How Human-like Behavior of Service Robot Affects Social Distance: A Mediation Model and Cross-Cultural Comparison"

_behavsci, 2022, doi:10.3390/bs12070205_

Round 1

Reviewer 1 Report

This is a very interesting well written study. No high-level comments, just a few following suggestions/ questions.

  1. Abstract – line 25 - Please add a statement at the end, about how these recommendations will impact future studies or future development of service robots.
  2. Line 46-48 – Please add citation.
  3. Lines 49-50 – Has HLBR been studied before?
  4. The authors explained the research gap and the intention for their research very well in the introduction.
  5. The authors selected participants based on their country of origin in a non-random pattern which introduces selection bias, which may skew the results. Please mention this in the study limitations.
  6. I would recommend separating out your discussion and conclusion sections. Please provide a concise conclusion at the end of the study.

Author Response

Response to Reviewer 1 Comments

Point 1: Abstract - line 25 - Please add a statement at the end, about how these recommendations will impact future studies or future development of service robots.

Response 1: We apologize for our lack of the statement of the practical significance of the study. We have added the significance of the study after the Abstract. We state, “This study provides suggestions for the human-likeness design of service robots to promote natural interaction between humans and service robots and increase human acceptance of service robots.”

Point 2: Line 46-48- Please add citation.

Response 2: We apologize for our lack of the citation. We add, “For example, the designer would make a robot's face look like a human's or add more human characteristics to its actions [10].”

Point 3: Lines 49-50 - Has HLBR been studied before?

Response 3: We apologize for not citing previous studies on HLBR. HLBR has been studied before. We have added the citation, “A few previous empirical studies have explored the human-like behavior of the service robot (HLBR) [8].”

Point 4: The authors explained the research gap and the intention for their research very well in the introduction.

Response 4: Thank you!

Point 5: The authors selected participants based on their country of origin in a non-random pattern which introduces selection bias, which may skew the results. Please mention this in the study limitations.

Response 5: Thank you for pointing this out to us. We apologize for the lack of this research limitation. We have added the limitation of the study after the Limitations and future research.

We state, “Fourth, we selected participants based on their country of origin; therefore, selection bias may have affected the experimental results. Future research could improve the selection method.”

Point 6: l would recommend separating out your discussion and conclusion sections. Please provide a concise conclusion at the end of the study.

Response 6: Thank you for your recommendation. We have separated the Discussion section and Conclusion. Conclusion have revised as follows:

  1. Conclusion

This paper aimed to explore the effects of HLBR on SDHR. Based on social identity theory, we constructed a conceptual model with perceived competence and perceived warmth as mediating variables. We recruited participants from China (219) and the United States (180) to complete the online experiment. Through empirical testing, we found that high HLBR (vs. low) led to higher perceived competence and perceived warmth, which shortened SDHR.

Reviewer 2 Report

This study examined factors influencing human acceptance of the service robot by surveying human awareness and attitude to service robots that imitate human appearance and behavior.In addition, they also looked at the difference in perceptions of robots in the US and China, which have completely different backgrounds in human relations.

It was quite interesting topic, but I felt the need to modify or add some parts to reinforce logical development as follows.  

1. Purpose and the current usage status of service robots will help understand the research purpose. Concretely, What are robots made for?, What are service robots used for?, What is the reason that human-like characteristics are required for service robots?, How are service robots currently being used?, Are there any issues pointed out there?, and Is "human-like characteristics" necessary to be accepted by consumers or people?

2. Please explain the characteristics of human relations, and usage states of robots (including service robots) in U.S. and China, in the introduction.

3. Regarding the survey participants, if you have collected opinions about the participants' direct or indirect experiences with robots, please summarize them in the results. 

4. Did you inform the information of the role of the research target, the service robot, to the survey participants before conducting the survey?

5. Line 124 “Hypothesis 5 (H5). Perceived warmth is negatively related to SDHR.” SDHR

If you want to explain the intimacy of social distance, it is better to avoid the use of "negative".

Author Response

Response to Reviewer 2 Comments

Point 1: Purpose and the current usage status of service robots will help understand the research purpose. Concretely, What are robots made for? What are service robots used for? , What is the reason that human-like characteristics are required for service robots?, How are service robot scurrently being used?, Are there any issues pointed out there?, and Is "human-like characteristics" necessary to be accepted by consumers or people?

Response 1: We apologize for the lack of explanation of the current use of service robots and insufficient research purposes. We have added a description of this section at the beginning of the introduction. We state, “Robots can be used to perform a series of complex actions [1]. A service robot performs service tasks for humans or devices [2]. It is an autonomous robot capable of interacting with people and completing specific service tasks [1]. The development of artificial intelligence technology has popularized service robots, such as educational robots, therapeutic robots, and entertainment robots [3]. However, human acceptance of service robots is the main obstacle to popularizing service robots [4]. Since service robots have certain social attributes [5] and human-like characteristics that can encourage humans to treat service robots as social participants, human-like characteristics can in-fluence the service effectiveness of robots [5,6]. The human-like characteristics of robots can effectively influence human attitudes toward robots [6]. Human acceptance of human-like characteristics of robots promotes human acceptance of service robots [7], whereas human non-acceptance of human-like characteristics of robots inhibits human acceptance of service robots [6].”

Point 2: Please explain the characteristics of human relations, and usage states of robots (including service robots) in U.S. and China, in the introduction.

Response 2: We apologize for our lack of the explanations on the interpersonal characteristics of China and the USA and the use of service robots. We have added this section at the end of the introduction. We add, “In the USA and China, robots are widely used in various fields, including the service industry [30], such as Sony's entertainment robot AIBO and Takara's home care robot TERA [27]. However, the two countries differ in their views on robots. Americans re-gard robots as assistants, while Chinese tend to regard robots as friends [31]. It is gen-erally believed that the USA is an individualist country and China is a collectivist country [32,33]. Compared to individualism, interpersonal relationships are more in-timate in the context of collectivism [34]. Social rules in interpersonal communication can also apply to human-robot interaction [22]. The cross-cultural study of human-robot interaction found that Chinese people have a higher sense of intimacy with robots than Americans [27].”

Point 3: Regarding the survey participants, if you have collected opinions about the participants' director indirect experiences with robots, please summarize them in the results.

Response 3: We apologize for not explaining this. We did not collect participants' experience comments.

Point 4: Did you inform the information of the role of the research target, the service robot, to the survey participants before conducting the survey?

Response 4: We apologize for not explaining this. We did not inform the participants the role information of the service robot. However, the robot introduced its own work (reception) in the video.

Point 5: Line 124 "Hypothesis 5(H5). Perceived warmth is negatively related to SDHR."SDHR

lf you want to explain the intimacy of social distance, it is better to avoid the use of "negative".

Response 5: Thank you for pointing this out to us. We apologize for our inappropriate expression. Expression have revised as follows:

Hypothesis 1 (H1). The higher the HLBR, the smaller the SDHR.

Hypothesis 2 (H2). The higher the HLBR, the stronger the perceived competence.

Hypothesis 3 (H3). The higher the HLBR, the stronger the perceived warmth.

Hypothesis 4 (H4). The stronger the perceived competence, the smaller the SDHR.

Hypothesis 5 (H5). The stronger the perceived warmth, the smaller the SDHR.